# Genome-Wide Identification and Expression Profile Analysis of Sugars Will Eventually Be Exported Transporter (*SWEET*) Genes in *Zantedeschia elliottiana* and Their Responsiveness to *Pectobacterium carotovora* subspecies *Carotovora* (*Pcc*) Infection

**DOI:** 10.3390/ijms25042004

**Published:** 2024-02-07

**Authors:** Ziwei Li, Yanbing Guo, Shoulin Jin, Hongzhi Wu

**Affiliations:** 1College of Animal Science and Technology, Yunnan Agricultural University, Kunming 650201, China; diwuzisang@163.com; 2College of Horticulture and Landscape, Yunnan Agricultural University, Kunming 650201, China; yanbingguo@139.com; 3College of Agriculture and Biotechnology, Yunnan Agricultural University, Kunming 650201, China; jslwhz@163.com

**Keywords:** *Zantedeschia elliottiana*, *SWEET*, expression pattern, subcellular localization, *Pectobacterium* spp.

## Abstract

SWEET, sugars will eventually be exported transporter, is a novel class of sugar transporter proteins that can transport sugars across membranes down a concentration gradient. It plays a key role in plant photosynthetic assimilates, phloem loading, nectar secretion from nectar glands, seed grouting, pollen development, pathogen interactions, and adversity regulation, and has received widespread attention in recent years. To date, systematic analysis of the *SWEET* family in *Zantedeschia* has not been documented, although the genome has been reported in *Zantedeschia elliottiana*. In this study, 19 *ZeSWEET* genes were genome-wide identified in *Z. elliottiana*, and unevenly located in 10 chromosomes. They were further clustered into four clades by a phylogenetic tree, and almost every clade has its own unique motifs. Synthetic analysis confirmed two pairs of segmental duplication events of *ZeSWEET* genes. Heatmaps of tissue-specific and *Pectobacterium carotovora* subsp. *Carotovora* (*Pcc*) infection showed that *ZeSWEET* genes had different expression patterns, so *SWEETs* may play widely varying roles in development and stress tolerance in *Zantedeschia*. Moreover, quantitative reverse transcription-PCR (qRT-PCR) analysis revealed that some of the *ZeSWEETs* responded to *Pcc* infection, among which eight genes were significantly upregulated and six genes were significantly downregulated, revealing their potential functions in response to *Pcc* infection. The promoter sequences of *ZeSWEETs* contained 51 different types of the 1380 cis-regulatory elements, and each *ZeSWEET* gene contained at least two phytohormone responsive elements and one stress response element. In addition, a subcellular localization study indicated that *ZeSWEET07* and *ZeSWEET18* were found to be localized to the plasma membrane. These findings provide insights into the characteristics of *SWEET* genes and contribute to future studies on the functional characteristics of *ZeSWEET* genes, and then improve *Pcc* infection tolerance in *Zantedeschia* through molecular breeding.

## 1. Introduction

As one of three families of sugar transporters that have been identified in plants, sugars will eventually be exported transporter (SWEET) proteins are membrane-localized unidirectional transporters that transport sugar in an energy-independent manner to the cell membrane. They have a wider range of substrate types and are ubiquitous in plants, animals, and prokaryotes [1]. The SWEET protein family members are featured by the MtN3/Saliva motif (PF03083) and seven transmembrane helices (TMHs) [2]. In general, the *SWEET* genes are clustered into four clades. Genes which belong to the same clade have similar structures and functions. The primary function is to transport hexose sugars including fructose and glucose in Clade I and Clade II [3,4], transport sucrose efficiently in Clade III [5,6], and export fructose located on tonoplast membrane in Clade IV [3].

The *SWEETs* gene family was initially identified in *Arabidopsis thaliana* [3]. To date, with a large number of plant genomes sequenced, the systematic genome-wide identification of *SWEET* genes has been reported in many crops, vegetables, fruits, and flowers, such as *Oryza sativa* [7], *Zea mays* [8], *Glycine max* [9], *Brassica oleracea* [10], *Solanum lycopersicum* [11], *Solanum tuberosum* [12], *Cucumis sativus* [13], *Musa acuminate* [14], *Malus domestica* [15], *Litchi chinensis* [16], *Vitis vinifera* [17], *Rosa chinensis* [18], *Dendrobium chrysotoxum* [19], *Hemerocallis fulva* [20], and *Petunia axillaris* [21]. Previous studies have shown that *SWEET* genes are involved in multiple physiological processes related to phloem loading [22,23,24,25], nectar secretion [6], pollen/flower development [26,27,28], seed/fruit development [29,30,31,32,33,34], senescence [35,36,37], modulating gibberellins response [38,39], and abiotic stress response [40,41,42,43]. Furthermore, the *SWEET* genes play a crucial role in plant–pathogen interactions and have been shown to be targets of extracellular pathogens [3]. The induction of *SWEET* genes upon pathogen infection has also been reported in *A. thaliana* [44], *Oryza sativa* [45,46,47,48,49,50], *Lotus japonicus* [51], *Gossypium* spp. [52], *Medicago truncatula* [53,54], *B. oleracea* [10], *Ipomoea batatas* [55], *S. lycopersicum* [56], *Manihot esculenta Crantza* [57], and *V. vinifera* [17]. For example, the overexpression of *IbSWEET10* increases tolerance to *Fusarium oxysporum* infection in *I. batatas* [55]. *AtSWEET4/15/17* [4], *SISWEET15* [56], and *VvSWEET4* [17] were shown to be associated with resistance to *Botrytis cinerea.* In *Citrus reticulata*, the overexpression of *CsSWEET1* could promote the propagation of citrus canker [58]. In rice, the knockout/overexpression of *OsSWEET13* could modify bacterial blight resistance and susceptibility [59].

Although the *SWEET* genes have been demonstrated to play crucial roles in plant growth and plant–pathogen interactions, they have not been studied in *Zantedeschia*, especially the chromosome-level genome of *Zantedeschia elliottiana* that has been recently resolved [60]. *Zantedeschia* is an herbaceous perennial bulb flower, belonging to the genus *Zantedeschia* in the family Araceae, which is native to the swampy, woodland edge, or mountainous regions of southern Africa [61]. It is an economically important horticultural crop widely used in flower arrangement, pot production, flower baskets, garden use, and landscaping for its attractive appearance and long postharvest life with ornamental value. The genus *Zantedeschia* consists of eight species in two sections: section *Zantedeschia* including two species (*Z. aethiopica* Spreng. and *Z. odorata* Perry.) with a rhizomatous storage organ and white flower, which are evergreen, and section *Aestivae* including six species (*Z. rehmannii* Engl., *Z. jucunda* Letty., *Z. elliottiana* Engl., *Z. pentlandii* Wittm., *Z. valida* Singh., and *Z. albomaculata* Baill) with a tuberous storage organ and colorful flower, which are winter-dormant [61,62]. *Z. aethiopica* in section *Zantedeschia* has a strong tolerance to bacterial soft rot [63]. However, the colored *Zantedeschia* species or hybrids in section *Aestivae* easily contract soft rot caused by *Pectobacterium carotovora* subsp. *Carotovora* (*Pcc*) [64,65]. *Pcc* is a rod-shaped, flagellated, parthenogenetic, anaerobic Gram-negative bacterium, of which the growth temperature is 4–36 °C, the optimum is 25–30 °C, and the lethal is 50 °C [66]. It is intolerant to dryness and sunlight, although commonly found in soil and water, and can only survive for about 15 days when left alone in the soil separate from the host [66]. The bacteria can infect any part of the calla lily plant under favorable conditions, especially during the stress period with high temperature, high relative humidity, high nitrogen content, and high moisture of the soil [65]. The pathogen can cause water-soaked spots in the early stage of infestation of *Zantedeschia*. With the rapid expansion of the spots, the virus may cause discoloration and depression, which will eventually lead to the plant rotting and even dying, greatly hindering the development of their industry and leading to substantial economic loss [65,66]. Therefore, it is urgent to breed new varieties of *Zantedeschia* that are resistant to soft rot disease. In this study, we have conducted the first genome-wide analysis of *SWEET* genes in *Z. elliottiana*, named *ZeSWEET* genes, and analyzed their chromosomal distribution, gene structure, motif composition, phylogenetic relationships, cis-regulatory elements, tissue-specific expression patterns, and expression in response to *Pcc* infection. The present study provides insights for future research on *ZeSWEET* genes associated with growth and development, and the response of *SWEET* genes in *Zantedeschia* to *Pcc* infection. Our results provide a solid foundation for the deep mining of genetic variations and potential genes involved in genes resistant to bacterial sot rot in *Zantedeschia* spp. and pave the way for efficiently breeding new *Zantedeschia* cultivars by a molecular breeding strategy.

## 2. Results

### 2.1. Genome-Wide Identification of SWEET Family Genes in Z. elliottiana

Based on the sequenced genomes of *Z. elliottiana* and combined with the homology search and conservative domain analysis, 19 non-redundant *ZeSWEET* genes were confirmed. We designated them *ZeSWEET1*–*ZeSWEET19* according to chromosomal location in this study for convenience. As shown in Figure 1, a total of 19 *ZeSWEETs* were unevenly distributed on 10 chromosomes. Among them, Chr14 contained the most *ZeSWEET* genes (four genes, accounting for 21.1%), Chr01 and Chr12 contained three *ZeSWEET* genes, Chr09 and Chr10 contained two *ZeSWEET* genes, while Chr04, Chr08, Chr13, and Chr16 contained only one *ZeSWEET* gene.

The ZeSWEET proteins were 97 (ZeSWEET12) to 870 (ZeSWEET17) amino acid residues long, with molecular masses between 10.77 (ZeSWEET12) and 96.94 kDa (ZeSWEET17), and the theoretical isoelectric points ranged from 6.52 (ZeSWEET17) to 9.85 (*ZeSWEET02*) (Table 1). Except for ZeSWEET17, the theoretical isoelectric points of the ZeSWEET proteins were greater than seven; thus, it was presumed that most ZeSWEET proteins were basic proteins. The instability index varied from 28.13 (ZeSWEET14) to 58.41 (ZeSWEET16), revealing that ZeSWEET proteins included stable and unstable proteins. The grand average of hydropathicity of four ZeSWEET proteins was less than zero, and others were greater than zero. So, it can be suggested that they have both hydrophobic and hydrophilic proteins. These results indicate that the basic properties of the proteins encoded by members of the calla lily *ZeSWEET* gene family were highly variable; therefore, these genes may play different functional roles in different biological processes.

### 2.2. Phylogenetic Analysis of SWEET Family Genes in Z. elliottiana

To define the evolutionary relationships among the ZeSWEET and other plant species, a phylogenetic tree was constructed by aligning 19 ZeSWEET protein sequences, 17 AtSWEET protein sequences, and 21 OsSWEET protein sequences. According to this phylogenetic tree (Figure 2), all SWEET proteins can be clustered into four clades, as previously reported for *A. thaliana* and *O. sativa* [7,8]. Clade I contains four ZeSWEET proteins (ZeSWEET04/07/13/17), clade II contains eleven (ZeSWEET01/02/03/05/06/08/09/10/11/12/16), clade III contains only three (ZeSWEET14/18/19), and the remaining one belongs to clade IV (ZeSWEET15). These results indicate that SWEET proteins are relatively conserved in plant evolution.

### 2.3. Gene Structure and Conserved Motif Analysis of SWEET Family Genes in Z. elliottiana

Structures and components of genes can render their sophistication and diversity of corresponded functions. For a deeper understanding the functional and structural characteristics of ZeSWEET, we used MEME (v.5.5.5) to analyze the motifs of each member.

A total of 19 amino acid motifs were identified in 19 ZeSWEET (Figure 3); ZeSWEET12 had the fewest amino acid sequences with two motifs, while ZeSWEET19 had the most with eleven; other genes contained three–ten motifs, five motifs (motifs 1/2/3/5/7) of which were present in all genes (except ZeSWEET04/05/06/08/10/11/12/16, which do not have them all), and all in the same order. Motif 18 was discovered only in Clade I, Motifs 9/11/14 were discovered only in Clade II, and Motif 6 was discovered only in Clade III, while Motif 8 was only not discovered in Clade II, and Motif 12 was only not discovered in Clade Ⅳ.

### 2.4. Promoter Region Analysis of SWEET Family Genes in Z. elliottiana

Promoter is a sequence of DNA that RNA polymerase recognizes and binds to in order to initiate transcription of a gene, usually located upstream of the gene. Upstream of the core promoter, there are usually specific DNA sequences, known as cis-acting elements, to which transcription factors bind to activate or repress gene transcription.

To further understand the transcriptional regulation and potential functions of *ZeSWEETs*, the cis-regulatory elements in the 2000 bp sequences upstream of the translation start sites were determined and analyzed. A total of 51 different types of the 1380 cis-regulatory elements were annotated (Figure 4). The promoters of the 19 *ZeSWEETs* contained at least 38 cis-elements (*ZeSWEET19*), while *ZeSWEET07* contained the most (122 cis-elements). These elements are classified into five categories, comprising core elements (CAAT-box, TATA-box), light response (elements involved in light responsiveness, light response element), growth and development regulation elements (seed-specific regulation, meristem expression, circadian control, and endosperm expression), stress response (low temperature, drought, wound, anaerobic induction, defense, and stress), and hormone response (auxin, abscisic acid, gibberellic acid, jasmonic acid, and salicylic acid). The most frequent elements were light response elements. Similarly, elements related to cell cycle regulation, endosperm-specific negative expression, maximal elicitor-mediated activation, and differentiation of the palisade mesophyll cells were only present in Clade II, and the element related to the protein binding site was only present in Clade III. Meanwhile, elements related to defense and stress responsiveness were absent from Clade III, and the element related to salicylic acid responsiveness was absent from Clade IV. Elements related to abscisic acid responsiveness, gibberellin responsiveness, anaerobic induction, drought inducibility elements, and low-temperature responsiveness occurred randomly in all clades. These data indicate that *ZeSWEET* clades may have specialization, and further analysis is necessary to understand the function of each *SWEET* gene.

Among the plant hormone response elements, a total of one hundred and seventy-three cis-acting elements were associated, and every *ZeSWEET* contained at least two hormone response elements. Eight elements of the TCA-element involved in salicylic acid response were detected in seven *ZeSWEETs*, ten elements of the TGA-element involved in auxin response were identified in eight *ZeSWEETs*, forty-five elements of the ABRE involved in ABA response were identified in the promoter regions of sixteen *ZeSWEET* members, and eighty-four elements of the CGTCA-motif and TGACG-motif involved in methyl jasmonate response were identified in all *ZeSWEETs*. Among the plant growth and development regulation elements, a total of fifty-nine cis-acting elements were associated, and every *ZeSWEET* contained at least one growth and development regulation element. Thirteen elements of CAT-box involved in meristem expression were identified in eight *ZeSWEETs*. These results suggest that the *ZeSWEET* genes might be broadly involved in growth and development, and hormone crosstalk in *Zantedeschia*. Among the plant stress response elements, a total of eighty-six cis-acting elements were associated, and every *ZeSWEET* contained at least one stress response element. Thirteen elements of LTR involved in low temperature responsiveness were identified in nine *ZeSWEETs*, ten elements of TC-rich repeats involved in defense and stress responsiveness were identified in nine *ZeSWEETs*, twenty-seven elements of MBS involved in drought inducibility were identified in fourteen *ZeSWEETs*, and twenty-eight elements of ARE involved in anaerobic induction were identified in fourteen *ZeSWEETs*. These data suggest that *ZeSWEETs* may be involved in the response to environmental stress through a complex mechanism, and each *ZeSWEET* gene can be induced by different environmental stresses.

### 2.5. Snyteny Analysis of SWEET Family Genes in Z. elliottiana

Gene duplication is the basis for the divergence of homologous gene functions and the main reason for the generation of new functional genes, which is an important driver of genome and species evolution [67]. In the present study, none pair of *ZeSWEET* genes was identified as a tandem duplication and two pairs of segmental duplication events were identified, comprising *ZeSWEET03/05* and *ZeSWEET14/18* (Figure 5).

Orthologs between species in a conserved order are known as syntenic, yet divergence between species can alter the species number and location of species genes [68]. Thus, syntenic analysis can reflect evolutionary relationships between species, reveal conserved core genes, and help explain trait differences caused by chromosomal variation. In this research, the syntenic relationships of the *ZeSWEET* family were analyzed by constructing five intergenomic collinear maps with *A. konjac*, *C. esculenta*, *P. stratiotes*, *P. pedatisecta,* and *S. polyrhiza*. There were 10, 10, 15, 11, and 12 pairs of homologous genes between *Z. elliottiana* with *A. konjac*, *C. esculenta*, *P. stratiotes*, *P. pedatisecta*, and *S. polyrhiza*, respectively (Figure 6). The results indicate that compared to *A. konjac*, *C. esculenta*, and *P. pedatisecta*, *Z. elliottiana*, *P. stratiotes*, and *S. polyrhiza* had higher homology, and *Z. elliottiana* was more closely related to the phylogeny and evolution of *P. stratiotes*.

### 2.6. Expression Profiles of SWEET Family Genes in Different Tissues in Z. elliottiana

To understand the roles of *ZeSWEETs* during plant growth and development, expression patterns of the 19 *ZeSWEETs* in different tissues were extracted, and 17 of 19 genes were detected (Figure 7). None of the genes were expressed in all tissues, but *ZeSWEET06/09* were not expressed in any of the tissues, suggesting that they may be expressed in highly specific tissues or under specific conditions. Seventeen expressed genes can be broadly classified into three categories based on their expression profiles. Category I, genes are expressed only in one tissue and marginally or not present in others: *ZeSWEET02/11/12* were only expressed in the stamen, *ZeSWEET08* in the style, *ZeSWEET* in the bulb, and *ZeSWEET16* in the spathe. Category II, genes are expressed only in a few tissues and marginally or not present in others: *ZeSWEET03* was only expressed in the root and stamen, *ZeSWEET04* in the pistil and bulb, *ZeSWEET05* in the style, spathe, and pistil, *ZeSWEET14* in the stamen and pistil, *ZeSWEET18/19* in the root and stamen. Category III, genes are only not expressed in one tissue and are more or less expressed in all others: *ZeSWEET01/15* were only not expressed in the leaf, *ZeSWEET07/17* not in the root, *ZeSWEET13* not in the pistil. Furthermore, except for individual genes, gene expression patterns in the same clade showed similarity. The genes of Clade I and Clade IV are expressed to a greater or lesser extent in almost all tissues; this suggests that they may be extensively involved in the growth and development of the calla lily. The genes of Clade II and Clade III are expressed only in individual tissues; this suggests that they are only involved in the growth and development of one part of the calla lily. In summary, these results suggest that all *ZeSWEETs* have varied expression in different tissues, which may contribute to the functional diversity of *ZeSWEETs*.

### 2.7. Expression Analysis of SWEET Family Genes after Pcc Infection in Z. elliottiana

To explore the possible roles of *ZeSWEETs* in a biotic stress response, the expression patterns of *ZeSWEETs* were analyzed by using the RNA-seq data of *Z. elliottiana* after *Pcc* infection, and 14 of 19 genes were detected (Figure 8). In general terms, 14 expressed genes can be classified into three categories based on their expression profiles. Category I, genes were significantly induced, and expression was consistently increased; includes *ZeSWEET07/08/13*. Category II, gene was induced and then repressed, with an increase in expression followed by a decrease; includes *ZeSWEET01/02/03/04/14/17*. Category III, gene was clearly repressed, and expression was consistently reduced; includes *ZeSWEET05/15/16/18/19*. However, the expressed genes had varied expression patterns even if they belong to the same clade. The genes of Clade I and Clade IV expressed approximate consistency; Clade I was induced and Clade IV was repressed. The genes of Clade II and Clade III were expressed variously. These results suggest that the *ZeSWEET* genes are sensitive to *Pcc* infestation and that there may be a interacting mechanism.

In order to better understand the response of *ZeSWEET* genes to *Pcc stress*, qRT-PCR was used to inspect the expression of individual genes for various times after *Pcc* infection, and 14 genes were found with detectable expression (Figure 9). Included among these, 6 of 14 genes were downregulated, while 8 of 14 were upregulated. Interestingly, the upward adjustments were not the same for the eight upregulated genes: *ZeSWEET03/04/08/13/14/17* showed the highest level of expression at 12 h, *ZeSWEET02* at 24 h, and *ZeSWEET07* at 36 h. In summary, *ZeSWEET* genes were induced or repressed by *Pcc* infection, indicating their indispensable regulation role in adapting the biotic stresses of *Zantedeschia*.

### 2.8. Subcellular Localization of SWEET Family Genes

The interior of a cell can be further divided into distinct cellular regions or organelles, known as subcellular. Mature proteins must be inside specific subcellular organelles to perform stable biological functions. Based on the WoLF PSORT (https://wolfpsort.hgc.jp/, accessed on 19 November 2023) projected analysis, subcellular localizations of the ZeSWEET06 were predicted on the cell wall, and ZeSWEET07/10/12/17/19 were on the chloroplast, while other ZeSWEETs were on the plasma membrane (Table 1). To validate the predictions, ZeSWEET07 and ZeSWEET18 were heterologously and transiently expressed in tobacco leaf epidermal cells as translational GFP fusion proteins, and both proteins were found to be localized to the plasma membrane (Figure 10). These results indicate that ZeSWEETs may have a membrane protein function.

## 3. Discussion

### 3.1. The Characteristics of Zantedeschia SWEETS

Soluble sugars have important roles in the regulation of plant metabolism, and growth and developmental processes [69,70,71]. The transport of soluble sugars between different cells, tissues, and organs is facilitated by different types of sugar transporter proteins (STs), which are the most central functional proteins in sugar transport and metabolism [71]. Depending on their substrate specificity, the major STs in plants are classified into three types: sucrose transporters (SUT/SUCs), monosaccharide transporters (MSTs), and sugars will eventually be exported transporters (SWEETs) [72,73,74,75].

The SWEET proteins were screened from unidentified membrane protein genes in the *Arabidopsis* Membrane Protein Database by the Fluorescence Resonance Energy Transfer (FRET) method [4]. They are a class of bi-directional sugar transport proteins that are not dependent on pH, and are present in a wide range of organisms, including protozoa, postbiotics, fungi, bacteria, archaea, higher eukaryotes, mammals, and higher vascular plants, demonstrating their importance in cellular organisms [1,7].

With the rapid development of sequencing technologies, *SWEET* gene families have been reported for a wide range of plants; the number of reported *SWEET* genes varies from 17 to 108, with 17 in *Arabidopsis* [4], 17 in grapevine [17], 21 in rice [7], and 108 in wheat [74]. In the current study, 19 *ZeSWEET* genes were identified in the *Z. elliottiana* genome, and they were clustered into four clades (Clades I to IV), which was consistent with the results for *Arabidopsis* [4]. Almost every clade has its own unique motif, although these were widely present in all members. These results implied that, while *SWEET* genes transport sugar, the substrates may differ between each clade.

### 3.2. The Functions of Zantedeschia SWEETS

As a family of sugar transporter protein genes newly discovered in recent years, *SWEETs* are highly homologous in sequence and structurally conserved, and function as bi-directional transporters of sugars [75]. They promote the diffusion of sucrose across the cell membrane down the concentration gradient into the extracellular body on cellular efflux, and also act as low-affinity glucose transporter proteins to regulate the uptake of glucose across the cell membrane [75]. They thereby exercise their functions, such as pollen development [26], nectar production [6], plant seed growth and development [29], plant senescence [35,36,37], abiotic stresses [14,40,41,42,43], ion transport [22,23,24,25], interactions with pathogenic bacteria, and reciprocal symbiosis with microorganisms [4]. In recent years, some other functions of *SWEET* in plants have been discovered: *MtSWEET1b* was induced to be expressed during the reciprocal symbiosis between alfalfa and *Arbuscular Mycorrhizal* to provide glucose to the *AM* [53], and *AtSWEET13* and *AtSWEET14* can transport gibberellins [4].

In this paper, the expression patterns of *ZeSWEET* genes were variations. *ZeSWEET02/11/12* were substantially expressed in the stamen, and *ZeSWEET10* in the bulb. On the contrary, *ZeSWEET13* was unexpressed in the pistil, and *ZeSWEET17* in the root. The changes in expression during organ development also suggest that different *ZeSWEET* genes function at different stages of organ development. Moreover, expression of *ZeSWEET06/09* was undetectable in any tissues/organs. This suggests that they may be expressed in highly specific tissues and/or under specific conditions.

### 3.3. The Promoters of Zantedeschia SWEETS

Growth and development in higher plants are the result of orderly gene expression in time and space. The regulation of plant gene expression is influenced by a variety of internal and external environments and is a multilevel regulatory system. Among these regulatory systems, transcriptional regulation of gene expression is a key aspect, and the regulation of promoters plays a very important role in the transcription process. Therefore, the study of the functional sequence of promoters has great significance for understanding gene expression regulation. In 1961, Monod and Jacob discovered promoters for the first time [76], and since then, molecular biologists have been studying the structure and functions of promoters. Promoters are indispensable components of exogenous expression vectors, which can drive the high-level expression of target proteins in recipient cells.

In this study, a wide variety of cis-acting elements have been identified in the promoters of *ZeSWEET* genes, and they are associated with growth and development regulation, light response, stress response, and hormone response, suggesting that *ZeSWEETs* are involved in complex signaling pathway regulation. The promoter sequences of *ZeSWEETs* contained cis-regulatory elements for nine types of phytohormone responsive and six types of environmental factors, including ABRE, CGTCA and TGACG motifs, GARE motif, P-box, TATC-box, TCA-element, TGA-element, ARE, GC-motif, LTR, MBS, and TC-rich repeats. Among these phytohormone responsive elements, the promoter region of *ZeSWEET18* contained the greatest number of components with 18, while *ZeSWEET02* contained the fewest with two. These results suggest that the *ZeSWEET* genes might be broadly involved in phytohormone responsive in *Zantedeschia*. Among the plant stress response elements, each *ZeSWEET* gene contained at least one cis-element (except *ZeSWEET02*), and nearly all genes contained ARE. These data suggest that *ZeSWEETs* may be involved in the response to environmental stress through a complex mechanism, and that each *ZeSWEET* gene can be induced by different environmental stresses.

### 3.4. The Response of Zantedeschia SWEETS after Pcc Interaction

An increasing body of evidence points to the fact that during the co-evolution of plants and pathogens, plant pathogens have evolved complex mechanisms to hijack plant developmentally regulated sugars for transport to themselves to sustain growth and reproduction [4]. *SWEET* genes are targeted by extracellular pathogens, and these causative organisms alter *SWEET* gene expression to obtain sugars needed for growth. For instance, *VvSWEET4* is a glucose transporter protein localized in the plasma membrane in grapevine, and oxidative burst and widespread cell death triggered by necrotrophic fungal infections can regulate *VvSWEET4* to promote sugar acquisition from plant cells [17]. *AtSWEET2* is a sugar transporter protein localized to the vesicle membrane that resists *Pythium* infection by limiting root nitrogen fixation [44]. *MeSWEET10* can be induced by *TAL20Xam668* from *Xanthomonas axonopodis* pv. *manihotis* secreted, which increases the non-specific transport of sugars, leading to an increase in sugar content in the cellular interstitial space, and thus increasing plant susceptibility [57]. *OsSWEET11* can interact with leaf blight bacterium (*Xanthomonas oryzae* pv. *oryzae*), and this molecular interactive mechanism increases sugar efflux, which is ultimately utilized by pathogens [29]. *GhSWEET10* can interact with *Xanthomonas citri* subsp. *malvacearum*, and silencing it significantly reduces the susceptibility of cotton to this pathogen [52].

In this study, *ZeSWEET* genes respond differently after *Pcc* infestation, 14 of 19 genes with detectable expression. Among them, *ZeSWEET03/07/08/13* were significantly induced, and *ZeSWEET03/04/08/13/14/17* showed the highest level of expression at 12 h, and *ZeSWEET01/02* at 24 h. On the contrary, *ZeSWEET05/15/16/18/19* were significantly suppressed, and *ZeSWEET15/16* show the lowest level of expression at 36 h, and *ZeSWEET18 at* 24 h. These results suggest that *ZeSWEET* may be involved in the interaction of the calla lily and *Pcc*. As to how they interact and which genes are involved in the interaction, this is to be studied in the next step.

### 3.5. The Prospects of Zantedeschia ZeSWEETs Research

This study provides a detailed investigation of the 19 *SWEET* genes which were identified for the first time in *Zantedeschia*, and it also raises a number of questions. For example, it examines whether Motif 6 is necessary for sucrose transport, since it was specifically present in Clade III. In addition, it is still unknown what type of sugar is transported by *ZeSWEET* members in different clades and the roles they play in responding to phytohormones. Thus, in future, with respect to *ZeSWEETs* in biotic and abiotic stresses, as well as plant growth and development, deep functional validation research by combining the transcriptome and metabolome is required to provide valuable insights to aid plant engineering strategies for developing crops resistance to adverse stress conditions. This early work on *SWEETs* in *Zantedeschia* will fundamentally assist in further molecular function characterizations and will help during explorations of genetic innovation.

As everyone knows, soft rot is a severe disease in *Zantedeschia* spp., especially all hybrids of section *Aestivae*, which are infected in varying degrees [63,64,65]. The breeding strategy of new calla lily varieties naturally revolves around this worldwide problem, which is to product a wide range of flower colors and high resistance to soft rot. At present, the strategy is mainly based on traditional breeding, i.e., crossbreeding [77]. Unfortunately, the plastomes and genomes of *Z. aethiopica* and cultivars of the section *Aestivae* were demonstrated to be incompatible to such a high degree that hybrids were chlorophyll deficient (albino) and could only survive heterotrophically [77,78], limiting the introgression of resistance genes from *Z. aethiopica* into section *Aestivae*. Compared with traditional breeding, transgenic breeding has incomparable advantages, mainly in breaking the barrier of reproductive isolation between species, bringing new genetic resources to species; targeting the traits of ornamental plants, making breeding more purposeful; and shortening the breeding cycle and time. Consequently, the most convenient and quickest way to investigate calla lily resistance is to find genes associated with soft rot. However, no successful disease-resistant transgenic varieties of section *Aestivae* have been reported currently in both domestic and foreign countries.

In our study, the RNA-seq data and qRT-PCR were used to explore the expression of *ZeSWEET* genes for various times after *Pcc* infection, and 14 of 19 genes were detected (Figure 8). Included among these, 6 of 14 genes were downregulated, while 8 of 14 were upregulated. Among the eight up-regulated genes, *ZeSWEET03/04/08/13/14/17* showed the highest level of expression at 12 h. These results suggest that *ZeSWEET* may be involved in *Pcc* infection in *Zantedeschia*, thus providing a potential genetic resource and research directions for molecular breeding for disease resistance in the calla lily.

## 4. Materials and Methods

### 4.1. Plant Materials and Stress Treatments

*Z. elliottiana* was planted in the base of College of Horticulture and Landscaping, Yunnan Agricultural University, Kunming, China (geospatial coordinates: 102.83945, 24.88627). The *Z. elliottiana* seedlings (with 2 leaves) under similar growth conditions were transplanted into square plastic flowerpots (10 cm long × 10 cm wide × 12 cm high, one plants per pot) containing peat and perlite (6:4 ratio) and placed in a light incubator at 22 °C, under 75% relative humidity and day and night cycles of 10/14 h. Three-month-old seedlings with 5–6 leaves were selected for the stress treatment experiments. To investigate the *Pectobacterium* pathogen responses of *Z. elliottiana* plants, the treatment group plants were inoculated with 10 µL of *Pcc* suspension (108 CFU/mL, OD600 = 0.1) and distilled water as a control. They were placed in a 28 °C incubator and leaves were collected at 0, 12, 24, and 36 h after treatment, respectively. The harvested samples were instantly frozen in liquid nitrogen and were kept at −80 °C until RNA extraction.

### 4.2. Identification of the SWEET Gene Family in Z. elliottiana

The genomic and annotative data for *Z. elliottiana* was obtained from Figshare (https://doi.org/10.6084/m9.figshare.22656112, accessed on 18 November 2023), and the candidate sequences of the *SWEET* genes of *A. thaliana* were downloaded from TAIR (https://www.arabidopsis.org/index.jsp, accessed on 18 November 2023). The hidden Markov model (HMM) profile of the *SWEET* (PF03083) gene family was obtained from Pfam (http://pfam.xfam.org, accessed on 18 November 2023). We used both the hidden Markov model (HMM) with default parameters and BLAST searches to search separately for *SWEET* genes in the *Z. elliottiana* with the “trusted cutoff and E-value < 0.01” as the threshold. *SWEETs* from *Z. elliottiana* were validated by conducting a BLAST search using *SWEETs* from Arabidopsis as queries. Then, the results of the two methods were intersected to obtain the candidate sequences, whose conserved domains were further identified by the Pfam database (http://pfam-legacy.xfam.org/null, accessed on 18 November 2023) and the Conserved Domain Database (CDD) (http://www.ncbi.nlm.nih.gov/Structure/cdd/wrpsb.cgi/, accessed on 18 November 2023). Sequences without a SWEET domain were removed. Nineteen ZeSWEET proteins in *Z. elliottiana* were finally obtained. Physicochemical characteristics and the subcellular localization of the ZeSWEET proteins were respectively predicted by the ProtParam tool (http://web.expasy.org/protparam, accessed on 19 November 2023) and WoLF PSORT II (https://wolfpsort.hgc.jp/, accessed on 19 November 2023).

### 4.3. Phylogenetic Analysis of SWEET Family Members

The homologous sequence alignment of the SWEET protein sequences in *A. thaliana*, *O. sativa*, and *Z. elliottiana* (Appendix A) were conducted using MAFFT (v.7.471) with default settings [79]. Among them, the data for *O. sativa* were obtained from NCBI (accessed on 19 November 2023). Based on the results of sequence alignment, a maximum-likelihood phylogenetic tree was constructed by MEGA7.0 with 1000 bootstrap replicates [80]. In addition, it was prepared (annotated and modified) by EvolVIEW v2.0 (https://www.evolgenius.info/evolview-v2/, accessed on 21 November 2023) [81].

### 4.4. Conserved Motifs, Conserved Domains, and Gene Structures Analysis

MEME software (v.5.5.5) (https://meme-suite.org/meme/tools/meme, accessed on 21 November 2023) was used to analyze the conserved motifs of ZeSWEET proteins. CDD (https://www.ncbi.nlm.nih.gov/cdd/, accessed on 21 November 2023) was used to identify the conserved domains of ZeSWEET proteins. The Gene Structure View (Anvanced) belonging to TBtools software (v.2.028) [82] was used to visualize the conserved motifs, conserved domains, and gene structures of ZeSWEET proteins and genes.

### 4.5. Chromosomal Localization and Colinear Analysis

The information on location for each *ZeSWEET* gene was obtained from the GFF genome annotation of Z. *elliottiana*. The chromosomal positions of *ZeSWEETs* were visualized using TBtools (v.2.028) [82].

One Step MCScanX from TBtools was used to identify the pattern of gene duplication with default parameters (E-value cut-off < 1 × 10^−10^ and Num of BlastHits with 5) and was used to analyze *SWEET* genes in Z. *elliottiana* vs. itself and Z. *elliottiana* vs. *Colcasia esculenta*/*Pistia stratiotes*/*Spirodela polyrhiza*/*Amorphophallus konjac*/*Pinellia pedatisecta*, *Colcasia esculenta*, respectively. The download sites of genomic and annotative files have been listed in Appendix A. The results were visualized using TBtools (v.2.028) [82].

### 4.6. Promoter Cis-Regulatory Element Analysis

The 2000 bp genomic DNA sequences upstream of the translation start sites of the *ZeSWEET* genes were searched for 5′ regulatory sequences using PlantCARE (http://bioinformatics.psb.ugent.be/webtools/plantcare/html/, accessed on 22 November 2023) to identify cis-acting elements. The prediction results of cis-acting elements were illustrated by TBtools (v.2.028) [82].

### 4.7. RNA-Seq Data Analysis

To explore the expression patterns of *ZeSWEET* genes in different tissues of *Z. elliottiana*, the transcriptome data, which include root, stem, leaf, style, stamen, spathe, pistil, and bulb, were downloaded from the NCBI SRA database (https://www.ncbi.nlm.nih.gov/sra/, accessed on 23 November 2023) [60]. The raw data have been listed in Appendix A. The expression levels of *ZeSWEET* genes in different tissues were examined according to FPKM (Fragments Per Kilobase of transcript per Million mapped reads), and the heatmap was drawn using TBtools (v.2.028) [82].

To investigate the *ZeSWEETs* expression profiles in response to *Pcc* stress (the stress experimental could be seen in Section 4.1), the transcriptome data was obtained from our laboratory (unpublished). The differentially expressed *ZeSWEETs* with over 2-fold changes in the transcriptome data were screened, Heatmaps of these *ZeSWEETs* were visualized using TBtools (v.2.028) with log 2-based FPKM values [82].

### 4.8. RNA Isolation and qRT-PCR Analysis

Total RNA of leaves inoculated with *Pcc* and control subjects was extracted using the Polysaccharide and Polyphenol Plant Rapid RNA Isolation Kit (Tiangen Biotech, Beijing, China) according to the manufacturer’s protocol, then the DNA in total RNA was removed with DNase I (Tiangen Biotech, Beijing, China). The concentration and purity were checked using the NanoDrop2000 spectrophotometer. First-strand cDNA synthesis with reverse transcriptase (Tiangen Biotech, Beijing, China) according to the manufacturer’s protocol using 1 μg of RNA template. The expression analysis was conducted using FastKing One Step qRT-PCR Kit (SYBR) (Tiangen Biotech, Beijing, China) and software (v.2.4.1) (Applied Biosystems, Leuven, Belgium). The actin gene was used as a reference in all experiments. Primers used for qRT-PCR are listed in Appendix A. All reactions were performed in triplicate, and fold change was calculated using the formula 2^−ΔΔCt^. Finally, the data were analyzed via DPS v.9.01 (Institute of Computing Technology, Chinese Academy of Sciences, Beijing, China) and Microsoft Office Excel v.2019 (Microsoft, Redmond, MA, USA), and Duncan’s test was employed to compare significant differences between treatments, and all data were the mean ± standard error of three biological and three technical replicates.

### 4.9. Subcellular Localization of ZeSWEET07 and ZeSWEET18

The ZeSWEET07 and ZeSWEET18 CDS sequence were cloned into the pEGOEP35S-H-GFP vector to generate pEGOEP35S-H-ZeSWEET07-GFP and pEGOEP35S-H-ZeSWEET18-GFP. The recombined plasmids were used to transform *Agrobacterium tumefaciens* GV3101 and were injected into leaves of *Nicotiana benthamiana* (4 weeks old). The infection solution was prepared with sterile water containing containing 10 mM MgCl_2_, 10 mM 2-Morpholinoethanesulfonic Acid (MES), and 0.1 mM Acetosyringone (AS), to an OD600 of 1.0. Two days after inoculation, the GFP fluorescence of these *N. benthamiana* leaves were observed by confocal laser scanning microscopy (Nikon A1 HD25, Tokyo, Japan) at 488 nm for GFP. The relevant primers are provided in Appendix A.

## 5. Conclusions

In this study, a total of 19 *SWEET* genes were identified in *Z. elliottiana* and were found to be distributed on ten chromosomes. They were clustered into four clades by phylogenetic analysis; each clade has its own specific motifs except for Clade Ⅳ, and the same branch had similar conserved motifs and gene structures. In the upstream sequence of the *ZeSWEETs* promoter, cis-acting elements related to light responsiveness, hormones, stress, and development were identified. The expression of *ZeSWEETs* is tissue-specific, which suggests the function diversity of these genes. Based on transcriptome data, the expression patterns of all the ZeSWEETs in *Pcc* infection indicate that they may be involved in *Pcc* stress signaling pathway regulation. Furthermore, qRT-PCR analysis showed that the expression of eight *ZeSWEET* genes increased and six *ZeSWEET* genes repressed when plants were exposed to *Pcc* infection. Overall, the results from this study provide a basic understanding of the *ZeSWEET* genes, facilitate unraveling the potential candidate *ZeSWEET* genes involved in the response to soft rot, provide valuable information to facilitate the breeding of soft rot-resistant cultivars in the calla lily, and provide a platform for the identification and comprehensive functional characterization of *SWEET* gene families from other Araceae plants.

## Figures and Tables

**Figure 1 ijms-25-02004-f001:**
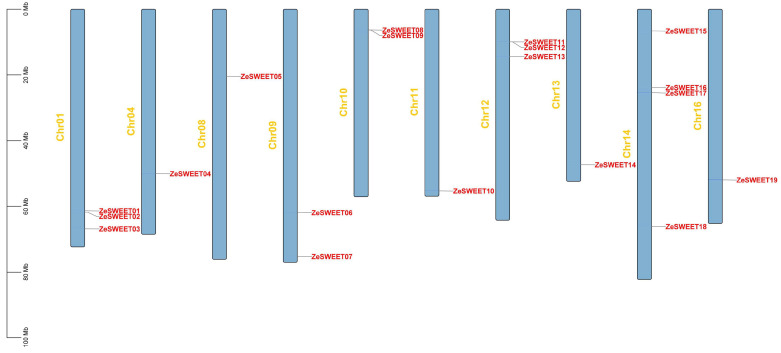
Distribution of *ZeSWEET* genes on the ten *Z. elliottiana* chromosomes. The bars represent chromosomes. The chromosome numbers are displayed on the left side, and the gene names are displayed on the right side. Each gene location is shown on the line. Detailed chromosomal location information is listed in Table 1.

**Figure 2 ijms-25-02004-f002:**
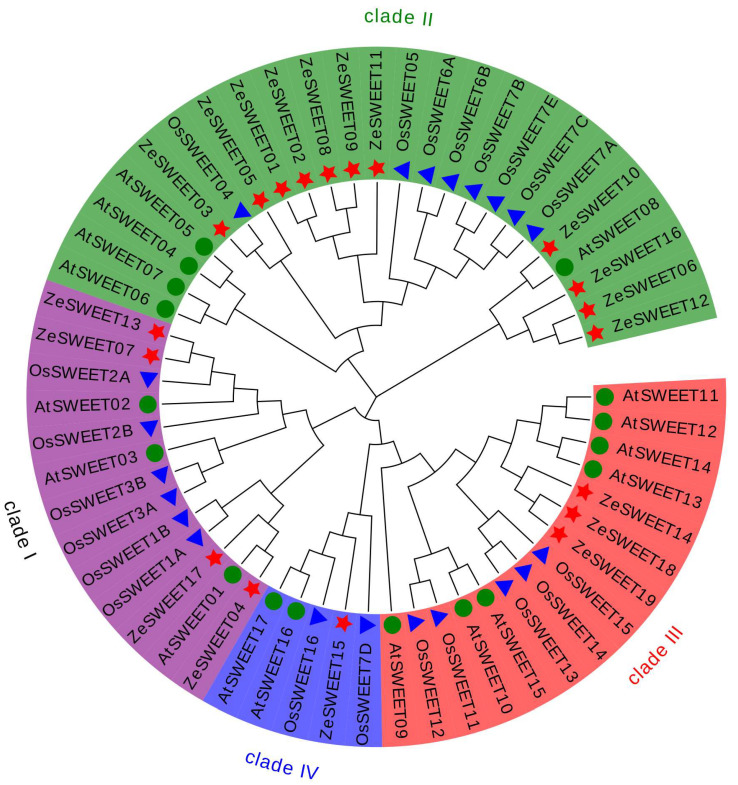
Phylogenetic tree for SWEET proteins of *Z. elliottiana*, *O*. *sativa*, and *A. thaliana*. Multiple sequence alignment of the SWEET domains was performed using MAFFT, and the phylogenetic tree was constructed using MEGA 7.0 with the maximum likelihood method with 1000 bootstrap replicates. The tree was divided into four clades, designated I, II, III, and IV. Different colors represent the four different clades: violet, Clade I; green, Clade II; red, Clade III; and blue, Clade IV. The blue triangle, green circles, and red star represent the 21 OsSWEETs in *O. sativa*, 17 AtSWEETs in *A. thaliana*, and 19 ZeSWEETs in *Z. elliottiana*, respectively.

**Figure 3 ijms-25-02004-f003:**
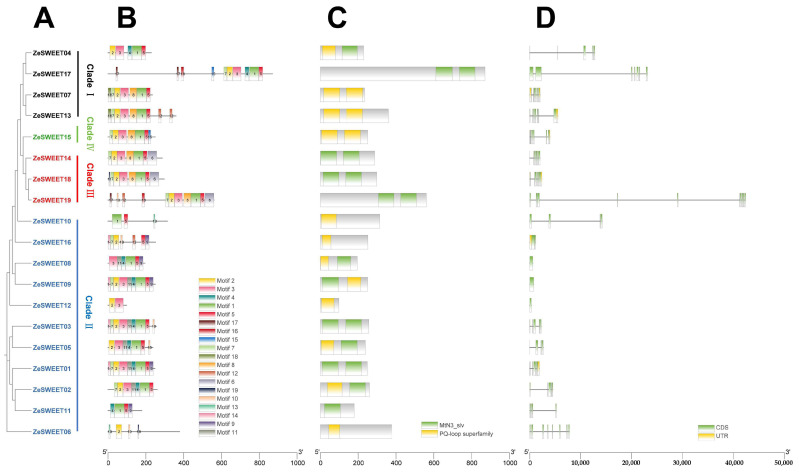
Phylogenetic relationship, gene structure, and conserved motif analysis of the *ZeSWEET* genes. (**A**) Phylogenetic tree of 19 ZeSWEET proteins. (**B**) Distributions of conserved motifs in ZeSWEET proteins. Nineteen putative motifs are indicated in different colored boxes. (**C**) The domain composition of ZeSWEETs. Green rectangles represent the MtN3/saliva domain and yellow rectangles represent the PQ-loop domain. (**D**) Exon-intron structure. Yellow boxes, green boxes, and black lines indicate the untranslated region, coding sequence, and gene length, respectively.

**Figure 4 ijms-25-02004-f004:**
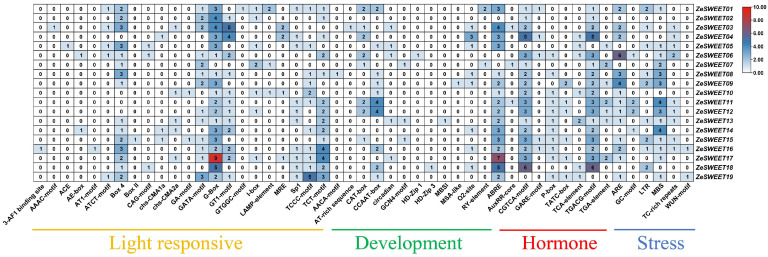
Cis-acting elements in the *ZeSWEET* gene family. Numbers of different elements in the promoter region of the *ZeSWEET* genes, as indicated by different color intensities and numbers in the grid.

**Figure 5 ijms-25-02004-f005:**
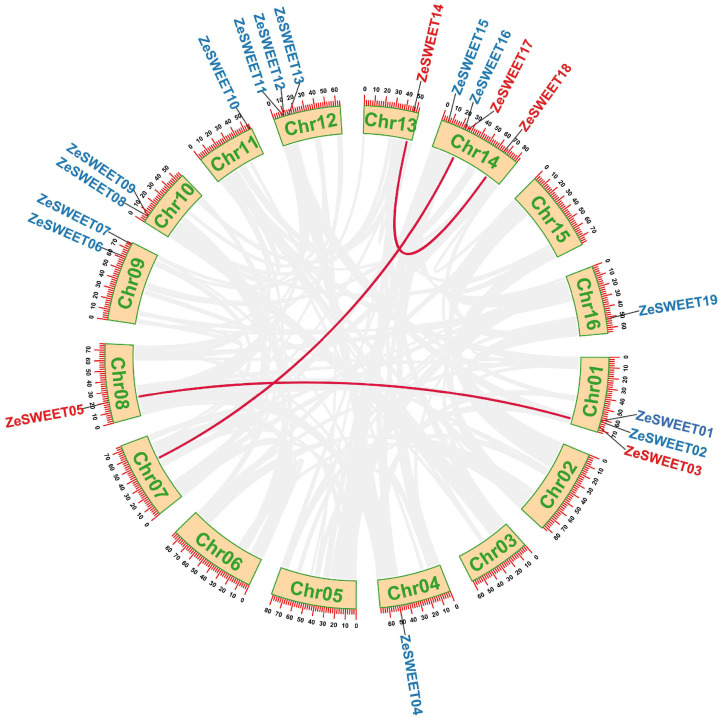
Gene duplication of the *ZeSWEET* genes. Red curves linking *ZeSWEET* genes indicate duplicated gene pairs in the *Z. elliottiana SWEET* family.

**Figure 6 ijms-25-02004-f006:**
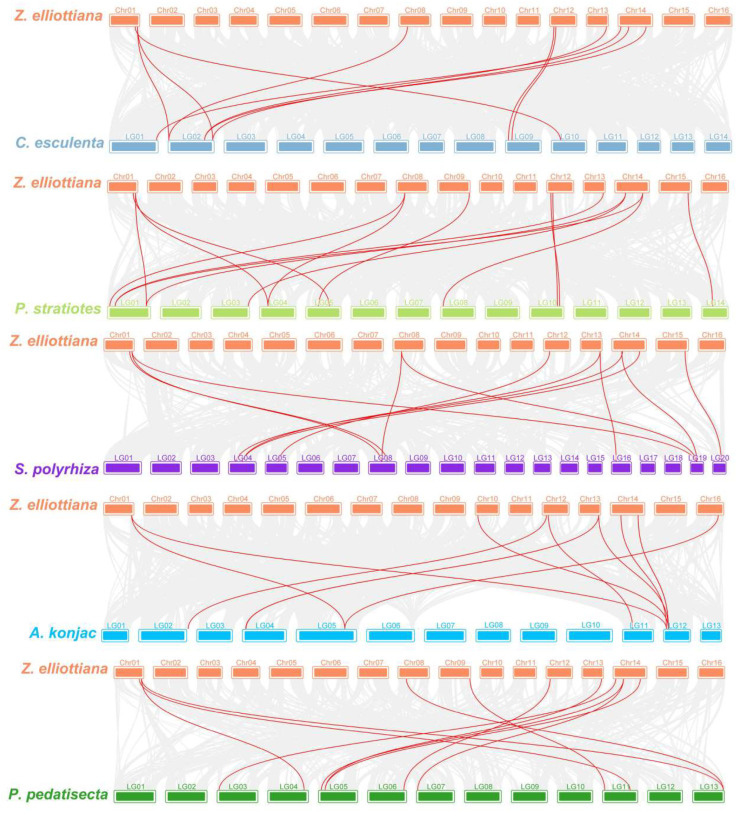
The collinearity relationship between the *Z. elliottiana* and other species. The collinear relationships between the *Z. elliottiana*, *A. konjac*, *C. esculenta*, *P. stratiotes*, *P. pedatisecta*, and *S. polyrhiza* genomes are shown on the chromosome maps. The gray line represents the collinearity between all members, and the red line represents the collinearity between the members of the *SWEET* family.

**Figure 7 ijms-25-02004-f007:**
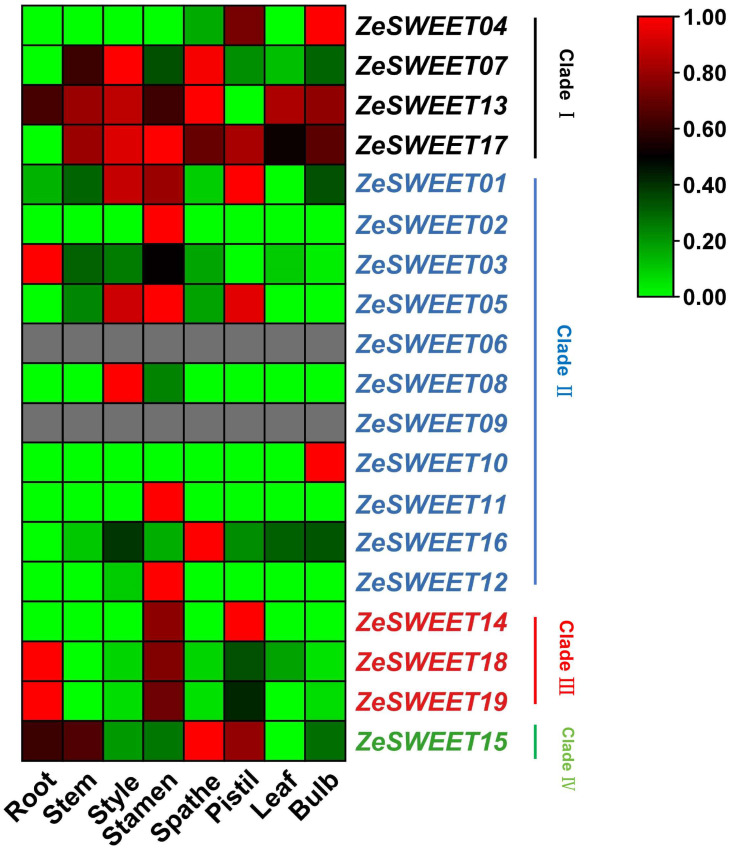
Expression patterns of *ZeSWEET* genes in different tissues analyzed by RNA-Seq. Red represents induced expression, green represents repressed expression, and gray indicates no expression (the same applies to Figure 8).

**Figure 8 ijms-25-02004-f008:**
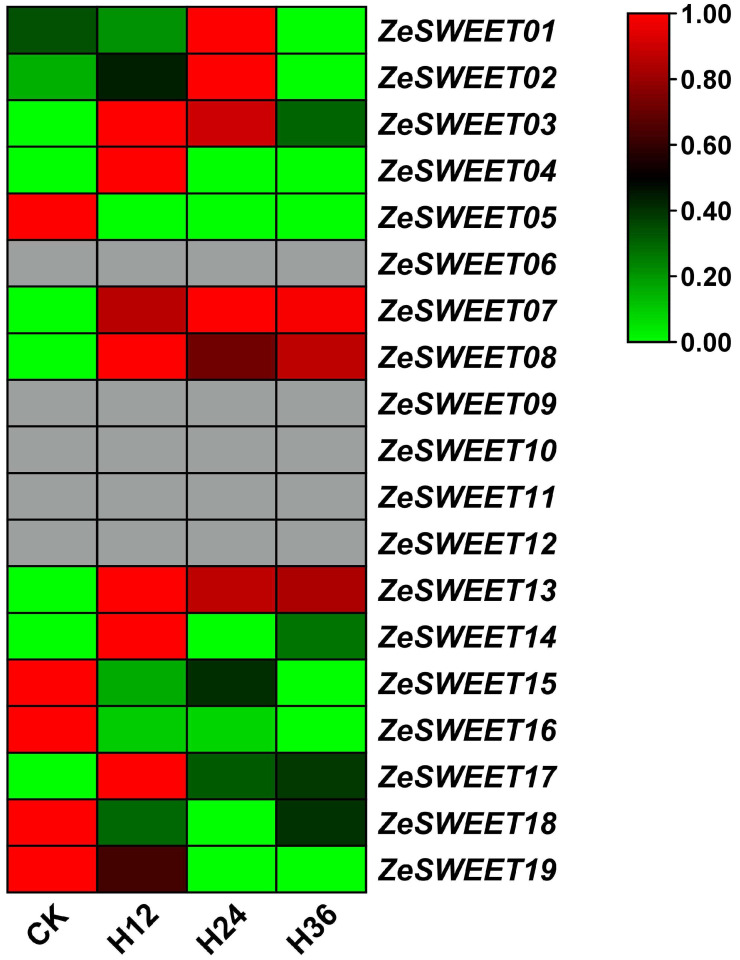
Expression patterns of *ZeSWEET* genes after *Pcc* infection analyzed by RNA-Seq. CK, control subjects. H, hours after *Pcc* infection (the same applies to Figure 9).

**Figure 9 ijms-25-02004-f009:**
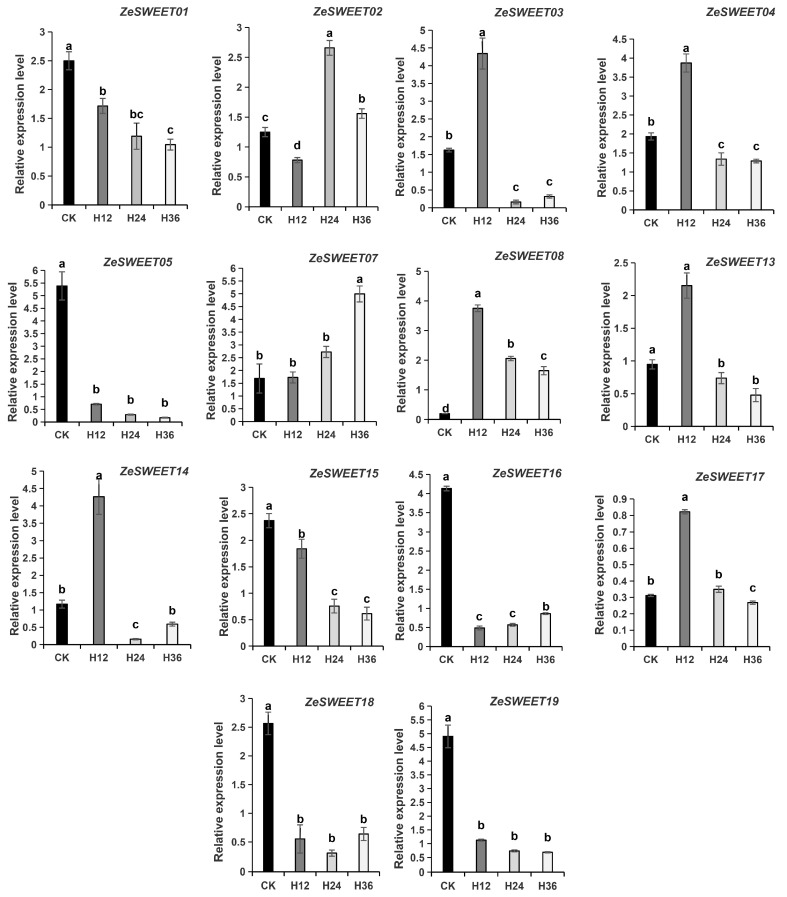
Expression levels of *ZeSWEET* genes in *Zantedeschia* leaves inoculated with *Pcc* via qRT-PCP. Data are presented as means ± standard deviations of three technical replicates derived from one bulked biological replicate. A Dunca’s multiple range test was used to calculate the significance level of the data at *p* < 0.05. Means denoted by the same letter are not significantly different at *p* < 0.05.

**Figure 10 ijms-25-02004-f010:**
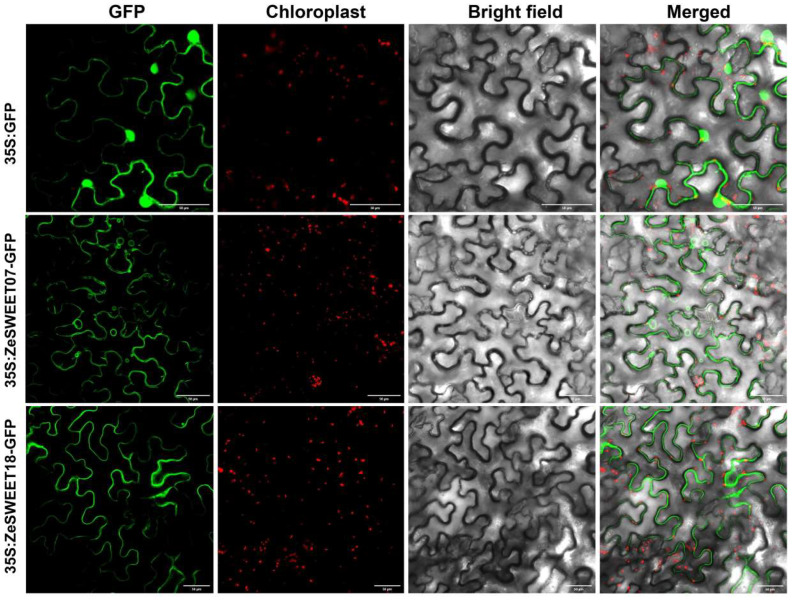
Subcellular localization of ZeSWEET proteins in tobacco leaves. From left to right are superposed photos of CK/ZeSWEET GFP green fluorescent protein, chloroplast spontaneous auto-fluorescence, bright field and merge (Scale bar = 50 μm).

**Table 1 ijms-25-02004-t001:** Characterization of ZeSWEETs in *Z. elliottiana*.

Gene ID	Location	Protein Length/aa	Molecular Weight/ku	pI	Instability Index	Aliphatic Index	Grand Average of Hydropathicity	Subcellular Localization
Zh01G133200.1	Chr01:61359146-61361011(+)	247	26,961.27	9.11	29.9	116.8	0.728	Plasma membrane
Zh01G134700.1	Chr01:61772271-61776799(−)	259	29,001.37	9.85	55.36	109.11	0.469	Plasma membrane
Zh01G158500.1	Chr01:66829835-66832087(−)	255	28,279.79	9.18	37.63	121.14	0.685	Plasma membrane
Zh04G140500.1	Chr04:50012720-50025486(−)	228	24,378.08	9.31	32.11	108.64	0.685	Plasma membrane
Zh08G102900.1	Chr08:20417532-20420150(+)	237	26,661.01	9.53	53.33	125.7	0.603	Plasma membrane
Zh09G140200.1	Chr09:61898091-61905886(+)	377	43,255.65	9.62	35.29	81.19	−0.324	Cell Wall
Zh09G204800.1	Chr09:75242560-75244590(−)	233	25,963.94	8.86	43.4	130.52	0.95	Chloroplast
Zh10G037900.1	Chr10:6326732-6327316(+)	194	20,842.97	8.63	42.12	127.11	0.881	Plasma membrane
Zh10G038000.1	Chr10:6337021-6337770(+)	249	26,936.27	9.21	35.67	123.25	0.808	Plasma membrane
Zh11G186300.1	Chr11:55237827-55252030(−)	313	34,379.46	8.64	52.63	85.4	−0.1	Chloroplast
Zh12G056300.1	Chr12:9955947-9961152(−)	178	18,862.06	8.5	38.8	101.91	0.471	Plasma membrane
Zh12G056400.1	Chr12:9961220-9961513(−)	97	10,767.91	9.21	36.43	121.65	0.601	Chloroplast
Zh12G077000.1	Chr12:14389758-14395284(+)	359	38,947.29	8.97	54.01	108.94	0.806	Plasma membrane
Zh13G136500.1	Chr13:47312228-47314257(+)	286	31,299.19	9.08	28.13	116.92	0.553	Plasma membrane
Zh14G035600.1	Chr14:47312228-47314257(+)	249	27,862.74	8.9	44.49	112.29	0.574	Plasma membrane
Zh14G134100.1	Chr14:23829044-23830197(−)	250	27,103.46	9.16	58.41	87	−0.03	Plasma membrane
Zh14G142500.1	Chr14:25381422-25404514(−)	870	96,943.95	6.52	50.44	89.94	−0.098	Chloroplast
Zh14G221800.1	Chr14:66140958-66143326(−)	297	33,004.09	9.17	35.95	114.61	0.582	Plasma membrane
Zh16G155400.1	Chr16:51874529-51916952(−)	560	62,646.7	7.2	57.6	98.95	0.137	Chloroplast

Note: pI, theoretical isoelectric point.

## Data Availability

The data presented in this study are available on request from the corresponding author.

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
