# Peer review of "Genome-Wide Identification and Expression Profile Analysis of Sugars Will Eventually Be Exported Transporter (*SWEET*) Genes in *Zantedeschia elliottiana* and Their Responsiveness to *Pectobacterium carotovora* subspecies *Carotovora* (*Pcc*) Infection"

_ijms, 2024, doi:10.3390/ijms25042004_

Round 1

Reviewer 1 Report

Comments and Suggestions for Authors

The authors have done a lot of molecular works to identify and analyze the 19 ZeSWEET genes in Zantedeschia elliotiana. Some of those observed genes had potential functions in response to Pectobacterium spp. infection. However, some issues need to be clarified or supplemented; thus, a minor revision is required. The comments in each section are included below.

Title

·       The title conveys the main topics of the study.

Abstract

·       The abstract is concise and can cover all the elements of the manuscript.

Keywords

·       The keywords are informative and cover the overall work of the study.

Introduction

·       Need a comma in line 41: As one of three families of sugar transporters have been identified in plants,Sugars….

·       Need a comma in line 45: mals,and prokaryotes……..

·       Line 51: “To data” ??

·       It is not a sentence in line 67-68. Please rewrite it to make it clearer.

…….. AtSWEET4/15/17 [4], SlSWEET15 [67], and VvSWEET4 [68] genes induced by Botrytis cinerea….

·       Line 72-75: Please rewrite it to make it clearer and understandable.

·       Line 81: ….two specie…..

·       Add more description about the Pectobacterium spp. including the life cycle, disease symptoms, and the statistical data to show the economic losses due to this pathogen.

Materials and Methods

·       Line 416: elliottiana….??

·       Line 418: under similar growth conditions….how many growing conditions applied in this study?

·       How to ensure that the Pectobacterium pathogen successfully penetrated and infected the plant samples?

·       How to ensure that the environmental conditions in the experiment supported the Pectobacterium pathogen to infect the plants?

·       Did the authors perform phenotyping on diseases incidence and or severity via scoring to validate the genotypic data? If yes, please provide it in this section.

Discussions

·       Provide a sub-section to discuss the implication of this study for conventional and molecular breeding aiming to develop resistant cultivars to soft rot disease.

Conclusions

·       Conclusions have addressed the research objectives.

References

            Please ensure that all citations are mentioned in the references.

Comments on the Quality of English Language

Minor editing of English language required.

Reviewer 2 Report

Comments and Suggestions for Authors

Authors have done good analysis of SWEET genes, however the implications of their findings and the how these will help in understanding these genes further needs to be described in more detail. 

The sampling of tissues for expression analysis is a little confusing and should be explained.

Provided figures/graphs can be elaborated for ease of understanding of the readers. 

Comments on the Quality of English Language

English language and sentence framing in some sections needs improvement 

Reviewer 3 Report

Comments and Suggestions for Authors

MS content is very good but written very poorly. Before the final publication of the manuscript, I would like to have some minor suggestions, and the authors need to consider these suggestions for quality work. Please look at the attached file, I have comments and suggestions

General Comments:

There are several flaws in structural grammar like commas and conjunctions are used frequently. The authors need to remove the unnecessary commas and conjunctions.

-Some sentences are redundant

-Some sentences are very long. I ask authors to rewrite the sentences to make them short for a clear message.

-Remove unnecessary words.

-At several places, there is a need for space between the words. 

- Authors needed to remove the repetition to improve the quality of the manuscript.

- Check the citation and reference style format.  

Reviewer 4 Report

Comments and Suggestions for Authors

The authors identified SWEET genes in Zantedeschia elliottiana based on homology search and looking for known conserved structural domains. They then seek to characterise the 19 SWEET genes they identify, including grouping them into four clades, analysing gene expression patterns in tissues and in response to infection, subcellular location and identifying cis-regulatory elements in their promoters. Based on their results, they suggest possible roles of these genes in the defence against plant pathogens.

In general, bioinformatics methods are based on well-known, established tools and they use qRT-PCR to support many of the expression patterns observed in response to infection. Given the little research available in this type of plant, the study provides interesting information for further study into these genes and their associated roles.

In general the captions could be expanded on in a number of the figures to make clearer what is shown - and please see specific comments below.

Figure 4: it would be of interest to compare to some background here - are these promoter elements more overrepresented in these genes compared to all genes in this species? Or are they typically found in all genes?

Figure 4: I checked the database for the promoter analysis (http://bioinformatics.psb.ugent.be/webtools/plantcare/html/) - it's from 2002 - surely there is a better, more recent source for cis-regulatory elements in plants that you can use in your study (probably also leading to more comprehensive results)?

Figure 4: Harmone > Hormone

Figure 8: Elaborate in the figure caption to say what CK, H12, H24, H36 refer to, what the grey represents and what the values (max 1 min 0) are referring to here. 

Figure 9: typically, one would use stars to represent where there are significant differences rather than letters - at least explain in the caption more clearly what these letters refer to. Say also here also what the tissue used for this analysis was.

What "No single gene is expressed in all tissues, but ZeSWEET06/09 were not expressed in all tissues which suggests that they may be expressed in highly specific tissues or under specific conditions." > "[..] were not expressed in any tissues, which suggests [..]"?

Comments on the Quality of English Language

I am a native speaker of English - overall, the English standard is high, but the paper could benefit from a read through due to small mistakes/grammatical issues, hence minor English corrections.
